# Hierarchically structured lithium titanate for ultrafast charging in long-life high capacity batteries

Mateusz Odziomek[1,2], Frédéric Chaput[1], Anna Rutkowska[3], Konrad Świerczek[3], Danuta Olszewska[3], Maciej Sitarz[2], Frédéric Lerouge[1] & Stephane Parola[1]

High-performance Li-ion batteries require materials with well-designed and controlled structures on nanometre and micrometre scales. Electrochemical properties can be enhanced by reducing crystallite size and by manipulating structure and morphology. Here we show a method for preparing hierarchically structured $Li_4Ti_5O_{12}$ yielding nano- and microstructure well-suited for use in lithium-ion batteries. Scalable glycothermal synthesis yields well-crystallized primary 4–8 nm nanoparticles, assembled into porous secondary particles. X-ray photoelectron spectroscopy reveals presence of $Ti^{+4}$ only; combined with chemical analysis showing lithium deficiency, this suggests oxygen non-stoichiometry. Electron microscopy confirms hierarchical morphology of the obtained material. Extended cycling tests in half cells demonstrates capacity of 170 mAh g$^{-1}$ and no sign of capacity fading after 1,000 cycles at 50C rate (charging completed in 72 s). The particular combination of nanostructure, microstructure and non-stoichiometry for the prepared lithium titanate is believed to underlie the observed electrochemical performance of material.

[1] Université de Lyon, Ecole Normale Supérieure de Lyon, CNRS UMR 5182, Université Lyon 1, Laboratoire de Chimie, 46 allée d'Italie, F69364 Lyon, France. [2] Faculty of Materials Science and Ceramics, Department of Chemistry of Silicates and Macromolecules, AGH University of Science and Technology, al. A. Mickiewicza 30, 30-059 Krakow, Poland. [3] Faculty of Energy and Fuels, Department of Hydrogen Energy, AGH University of Science and Technology, al. A. Mickiewicza 30, 30-059 Krakow, Poland. Correspondence and requests for materials should be addressed to F.C. (email: frederic.chaput@univ-lyon1.fr).

Ensuring effective ionic and electronic transport in the electrodes is crucial, to construct high-performance batteries. This issue is particularly important considering insertion- or intercalation-type electrodes utilized in lithium-ion batteries[1–4]. Depending on the intrinsic structural and transport properties of the particular electrode material, various approaches are used. For instance, if the material possesses insufficient ionic (that is, Li$^+$) conductivity, transition to nanoscale, as well as grain morphology optimization is performed, to decrease diffusion length or/and expose particular crystal planes, through which the enhanced ionic transport may occur[1,5–8]. At the same time, to increase electronic component of the conductivity, various additives (usually carbonaceous materials) are often used, yielding composite-type electrodes[9–11]. Alternatively, intrinsic transport properties, as well as chemical stability of the candidate electrode compound can also be adjusted, typically by appropriate chemical modification (such as doping)[8,12–14].

In the case of Li$_4$Ti$_5$O$_{12}$ (LTO) spinel, which is considered as a promising anode material for Li-ion batteries, especially for use against a high-voltage cathode, both ionic and electronic transport in the electrode must be optimized[8]. The material, which is particularly interesting due to its zero strain-like behaviour during lithium insertion/extraction[15], excellent chemical stability and good cyclability, unfortunately suffers from low ionic and electronic conductivities. Although inadequate transport properties of microcrystalline LTO hinder its usefulness, transition to nanoscale was shown to be an effective way to greatly improve electrochemical performance for this material[16,17].

Nanostructured LTO was previously reported in a form of nanosheets[18], nanorods[19], nanotubes[20], nanowires[21], nanoflakes[22] nanoflowers[23] and nanoparticles (NPs)[24]. The respective materials were prepared via various routes, including solid-state, hydrothermal, sol-gel, microwave, combustion, molten salt, sonochemical, rheological phase and spray pyrolysis methods (ref. 8 and references therein). Although nanostructured LTO particles can deliver superior rate performance, with substantial capacity attained at high current rates of 100C and higher, this usually comes with high irreversible capacity loss at initial cycles and with difficulty in manufacturing electrodes of high volumetric energy density, owing to low tap density of such powders[8]. As a solution, ideal hierarchically structured materials can be designed, in which primary, nanosized crystallites form larger, porous agglomerates, which can be further grouped into micro-sized grains. Well-controlled multiscale porosity would enable effective penetration of the liquid electrolyte, with distance for Li$^+$ diffusion minimized within the electrochemically active NPs. If at the same time, electronic transport in electrode can also be ensured (for example, by a carbon-based additive), LTO-based anodes having excellent electrochemical properties can be achieved.

Numerous reports of synthesis of nanostructured LTO can be found in the literature[15,25,26]. In many cases, the synthesis routes are complex and/or unsuitable for inexpensive commercialization, in particular if the observed performance improvements are incremental taking conventionally prepared LTO as a baseline. Promising syntheses of LTO have been reported using solvothermal approaches in various solvents, but to our knowledge never in 1,4-butanediol (1,4-BD), a solvent shown to exhibit particular properties for oxide synthesis[27].

In this work, we present synthesis of hierarchically structured LTO by a relatively inexpensive and scalable method. We use the

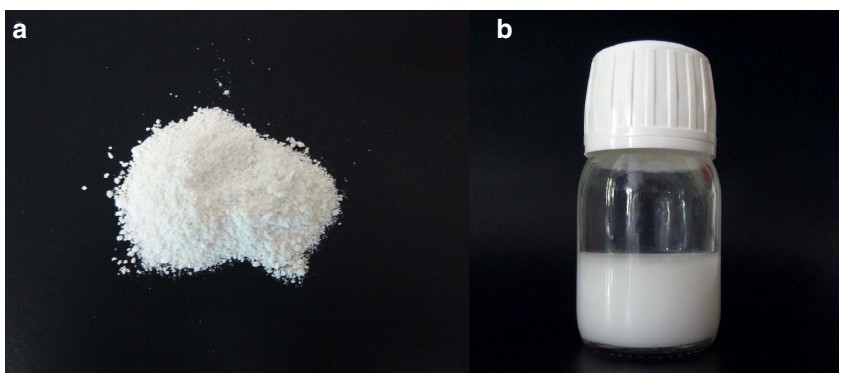

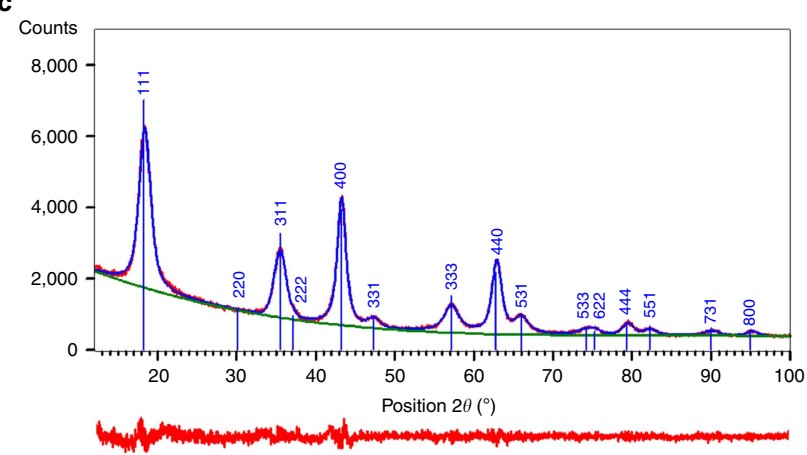

**Figure 1 | The obtained lithium titanate phase.** (**a,b**) Photographs of purified material in form of dried powder and in form of colloidal solution in ethanol. (**c**) XRD pattern (CuKα) performed on dried powder exhibiting broad peaks of pure lithium titanate phase due to the nanoparticulate character of the material. Red plot at the bottom represents the residual after fitting.

well-known glycothermal approach, in which oxide NPs are synthesized from metal-organic compounds in 1,4-BD in autoclaved conditions. Nanostructuring of the prepared LTO results in enhanced electrochemical performance in extended cycling tests at high rate.

## Results

**Lithium titanate NPs with hierarchical structure.** The synthesis was achieved by simple mixing of lithium acetate dihydrate and titanium sec-butoxide in 1,4-BD and subsequent heating at 300 °C for 2 h under autogenous pressure and under stirring. The obtained milky colloidal solution was centrifuged at 4,000 r.p.m. and washed with ethanol. The washing procedure was repeated three times to get pure crystalline powder after drying (Fig. 1a) or followed by re-dispersion of NPs in ethanol (Fig. 1b). During initial heating, a burst of nucleation occurred with generation of uniform primary NPs, which aggregated together creating spherical secondary particles of characteristic sizes of a few hundred nanometres. The particular structure of the as-prepared compound led to enhanced electrochemical performance, which was recorded on standard coin-type cells, without special or expensive modifications to routine electrode preparation methods.

According to X-ray powder diffraction (XRD) analysis (Fig. 1c), a highly crystalized pure LTO phase was obtained in agreement with the reported JCPDS data (Card Number 00-049-0207) and corresponding to a spinel type cubic structure. Significant peak broadening was ascribed to the small crystallite size. In general, such peak broadening can result from a number of factors such as instrumental issues, lattice distortions and small crystallite size. The last factor produces Lorentzian broadening, while the former two produce a Gaussian shape. The experimental XRD diffraction profile was better fitted with a Lorentzian function than Gaussian, implying that the crystallite size was the dominating factor. A Rietveld refinement was performed using data in the range 10°–100° in $2\theta$. The Thompson–Cox–Hastings formulation of the pseudo-Voigt function and an instrumental resolution function were applied to describe the peak profiles and to determine an apparent crystallite size, as well as the lattice parameters. The results led to an apparent crystallite size of 4.0 nm. The cell was refined in the cubic system, Fd-3m and the unit cell parameter $a = 8.364$ Å was slightly larger than values reported in the literature[28]. Interestingly, good crystallinity was observed without post treatments such as calcination.

Scanning electron microscopy (SEM), transmission electron microscopy (TEM) and electron diffraction studies (Fig. 2) confirmed the particular hierarchical nanostructure of the material. High-resolution TEM images show the formation of well-crystallized, cubic NPs free of secondary phases with typical sizes of 4–8 nm (Fig. 2a,b). This size is in accordance with the crystallite size calculated from XRD results. The ring electron diffraction pattern (Fig. 2b, inset) shows typical polycrystalline character of the sample. Spherical aggregates of the primary particles shown in the Fig. 2b have a diameter of several hundred nanometres and they are also observed in further electron microscopy pictures (Fig. 2c,d) as balls of similar diameters. Further, the aggregates of crystallites create even larger spheroidal structures with micrometric sizes. The overall hierarchical organization of the particles was observed using both SEM and TEM as shown by Fig. 2c,d. Formation of spherical aggregates of particles was previously observed for glycothermal synthesis of $Y_3Al_5O_{12}$ NPs where primary particles of the size of few tens of nanometres aggregated into secondary structures. Those primary particles could not be separated once aggregated. However, no larger scale aggregation was observed in that case[29].

Gas adsorption studies confirmed the porous structure of the material. Adsorption/desorption isotherms (Supplementary

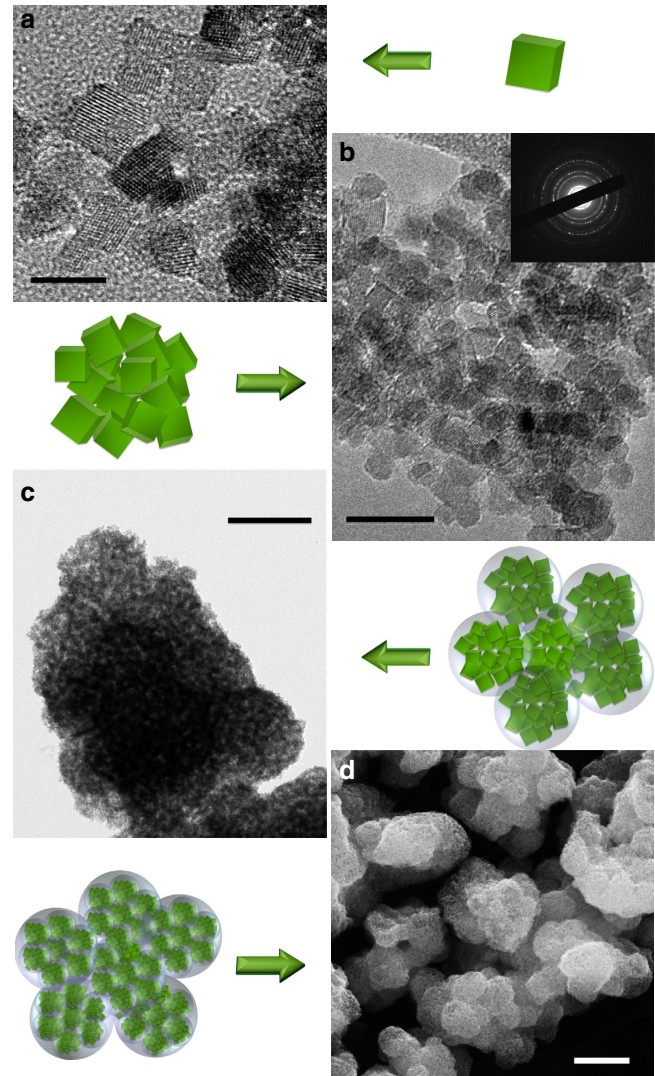

**Figure 2 | Hierarchical structure of the material.** (**a**) High-resolution TEM shows well-crystalized NPs. Scale bar, 10 nm. (**b**) High-magnification TEM image of single spherical aggregate. Scale bar, 20 nm. (**c**) TEM image of highly porous, sponge-like structure composed of spherical aggregates of fine particles. Scale bar, 500 nm. (**d**) SEM picture shows overall structure of spherical particles arranged into larger structures. Scale bar, 1 μm. The drawings schematically represent the observed three levels of particle arrangement with small NPs aggregated into spherical structures (150–500 nm size), which gather into even larger structures.

Fig. 1) corresponded to the isotherm type IV according to IUPAC nomenclature[30], which is characteristic for mesoporous materials. The hysteresis was due to the aggregations of NPs, which created voids in the mesoporous range, and thus resulting in capillary condensation. Calculated Brunauer–Emmett–Teller surface area was $220 \, m^2 \, g^{-1}$, which corresponded to cubic particles with diameter 7.8 nm. This surface area value is one of the highest reported in the literature for pure LTO phase. The pore size distribution calculated from Barrett–Joyner–Halenda model (Supplementary Fig. 1, inset), clearly showed three types of porosity: (i) coarser mesoporosity of 40–50 nm, (ii) finer mesoporosity of 2–4 nm and (iii) microporosity. This is consistent with the proposed multiple levels of assembly in the aggregates.

The surface of the as-prepared NPs was covered with organic species as shown by fourier transform infrared spectroscopy

(FTIR) (Supplementary Fig. 2) and thermogravimetric analysis (TGA) (Supplementary Fig. 3). The amount of organics was evaluated to be 12% by weight from TGA plot. This organic cover was attached by chemical bonds strong enough to remain after the washing step. X-ray photoelectron spectroscopy (XPS) and FTIR evidenced that the NPs were mostly covered with residual butoxy groups from titanium alkoxide and 1,4-BD.

**Stoichiometry deviation of nanostructured lithium titanate.** Interestingly, inductively coupled plasma optical emission spectroscopy (ICP-OES) analysis showed 4.64% of Li and 46.30% of Ti atoms, corresponding to a molar ratio Li/Ti = 0.692. Such significant deviation from the theoretical structure is surprising in light of the excess of lithium used during the synthesis. Calculations made from XPS also confirmed shortage of Li atoms (Supplementary Table 1 and Supplementary Fig. 4a). This gave rise to a ratio of 3.5 mol of Li to 5 mol of Ti in LTO. Charge neutrality considerations would suggest some $Ti^{3+}$ to be present in the structure. However, EPR measurement did not detect any $Ti^{3+}$. Moreover, the high-resolution Ti 2p core level XPS spectrum (Supplementary Fig. 4b) shows two peaks at 458.7 and 464.4 eV, which indicated the existence of $Ti^{4+}$ exclusively in the material. As a consequence, we assumed that the compound had some oxygen vacancies in the structure. However, there is a possibility that some charge is compensated by O–R groups (R = H or C) remained on the particles surface.

Such lithium deficiency, or in other words, formation of titanium-rich LTO can be explained on the basis of work by Lu et al.[31,32] where they investigated LTO structure by aberration-corrected scanning TEM. They noticed irregular surface layers (2 nm thick) with reduced contrast of lithium ion columns, in comparison with the interior regions. Authors concluded that the outmost layer possessed titanium-enriched composition. In addition, Wang et al.[33] observed rutile surface layer on synthesized LTO nanosheets when <0.5 mol excess of Li precursor was used. Consequently, it can be concluded that the surface structure of synthesized NPs undergoes a specific relaxation where titanium atoms are exposed on the surface. Therefore, higher ratio of titanium to lithium atoms in small NPs with comparable amount of surface and volume atoms is not surprising. This effect may also explain the presence of the oxygen vacancies in the synthesized LTO. Surface of grains of oxides experiences rearrangement of atoms usually yielding exposition of oxygen atoms. For instance, in a reconstructed surface structure of $SrTiO_3$ only $TiO_4$ units are present[34]. Considering that the near surface layer composition of the material is closer to $TiO_2$, which has lower Ti/O than LTO, oxygen deficiency can be expected. This explanation is supported by XPS measurement (which is more sensitive towards particle surface) where even more lithium deficiency was detected.

**Ultrafast charging with high capacity retention.** It is known that charge/discharge curves differ in shape for micro- and nanosized LTO, with flat voltage versus charge state observed for batteries made using microsized material and pronounced voltage changes when NPs are used[35–37]. Interestingly, the recorded curves, at low current rate of C/2, show a substantial capacity range (ca. 30–40 mAh g$^{-1}$) with voltage constantly changing with the ongoing electrode process. This is followed by a relatively flat region, and finally, larger changes at final stages of charge/discharge (Fig. 3a). The recorded capacity in 1.0–2.6 V range exceeds the theoretical one of LTO and is on the order of 190 mAh g$^{-1}$ after 10 cycles. Although on first discharge such behaviour might be interpreted as originating from initial introduction of missing lithium, speculatively into the 8a site with corresponding 29 mAh g$^{-1}$ of additional capacity, the

enhanced capacity on the following cycles is possibly linked with near-surface lithium storage, with contributions of the surface-present organics originating from the synthesis process[38,39]. In the next 100 cycles, capacity stabilized at around 170 mAh g$^{-1}$ (Supplementary Fig. 5).

Appropriately prepared, nanosized LTO spinel was already shown to exhibit excellent electrochemical properties, with optimized materials delivering almost theoretical capacity up to 50C, as well as capacities exceeding 100 mAh g$^{-1}$ at unusually high rates on the order of 500C or above[16,17]. However, in those works the cells were prepared using rather non-standard and therefore costly approaches. In addition, as shown in ref. 16, the active material loading was below 0.2 mg cm$^{-2}$, which is limiting from a practical point of view. As presented in the Methods section, the LTO-based electrodes were prepared by a standard method with a more typical loading of the active material of about 1 mg cm$^{-2}$.

Figure 3b presents the results of initial 1,000 cycles of CR2032 Li/Li$^+$/LTO battery recorded for 50C charge/discharge rate in 1.3–2.5 V range. Although the capacity in the initial cycles was low, it improved considerably on cycling, which can be directly related to the decreasing polarization (Fig. 3c). After about 300th cycles, the capacity stabilized close to 170 mAh g$^{-1}$ and, remarkably, no capacity fading was observed until the 1,000th cycle (Table 1). Although this degree of stability can be partially attributed to a relatively narrow cycling voltage range, it is mostly due to the intrinsic properties and morphology of the synthesized material. It should be also emphasized that metallic lithium anode maintained stable operation in those 1,000 cycles, despite known common problems with dendrite formation. Probably, good performance of Li/Li$^+$ electrode was related to the limited 1.3–2.5 V cycling range. Examination of cycled material microstructure by SEM revealed no visible differences. LTO remained highly porous and hierarchically structured (Supplementary Fig. 6). In addition, XRD did not show changes in peak broadening. Therefore, we conclude that the NPs did not undergo significant changes in shape or size.

Interestingly, a substantial discharge capacity on the order of 99 mAh g$^{-1}$ could be maintained if the electrode was subjected to an extremely fast 500C charge current and then discharged at 50C (Fig. 3d and Supplementary Fig. 7). This result is of practical interest, as it suggests that manufacturing of LTO-based batteries capable of ultrafast charging is attainable. Doubtlessly, such extraordinary results are connected with high ability of Li ions to be inserted or extracted into/from NPs with well-developed surfaces[39] and with the particular non-stoichiometry of the system. In addition, high crystallinity is also an important factor for good performance.

Representative results of cycling voltammetry studies, performed with 1 mV s$^{-1}$ scanning rate are shown in Supplementary Fig. 8. As can be noticed, there is a substantial asymmetry between the anodic and cathodic peaks, with the cathodic one being initially split in two. Origin of this effect is likely due to the ongoing solid-electrolyte interface (SEI) formation[40], which diminishes in the following cycles. This is also supported by electrochemical impedence spectroscopy data gathered on cycling, as presented in Supplementary Fig. 9. As expected, after SEI formation the recorded resistances decrease substantially and remain almost constant for the following cycles. Finally, the increase of capacity in the first 300 cycles at fast charging rate is ascribed here to rather complicated formation of SEI under a fast charging/discharging rate.

In summary, a method of preparation of hierarchically structured LTO-type nanosized spinel, based on scalable and facile glycothermal process, was developed. It was possible to obtain uniform, well-crystallized primary 4–8 nm NPs, self-assembled in

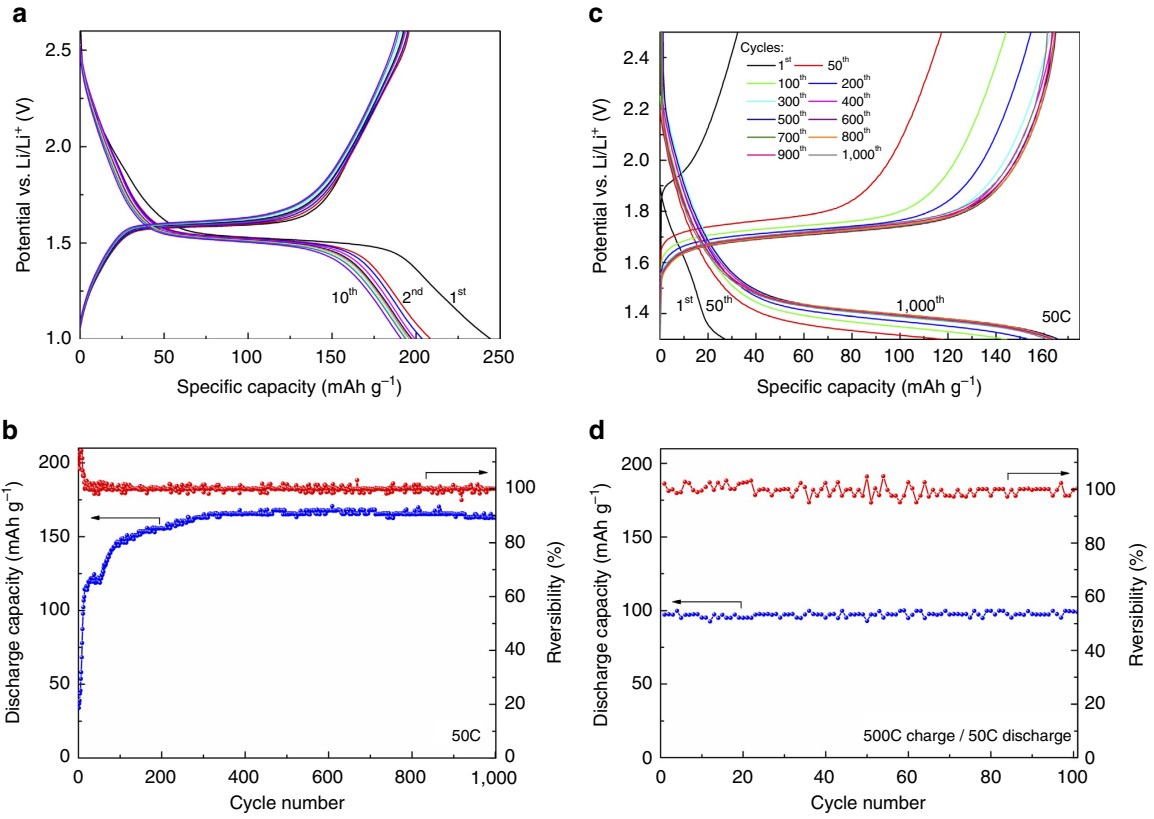

**Figure 3 | Electrochemical performance.** (**a**) Charge/discharge profiles for Li/Li$^+$/LTO battery cycled at C/2 current rate. (**b**) Initial 1,000 cycles of Li/Li$^+$/LTO battery at 50C current rate. The current density was 8.76 A g$^{-1}$. (**c**) selected charge/discharge profiles for Li/Li$^+$/LTO battery cycled at 50C current rate. (**d**) Discharge capacity of the Li/Li$^+$/LTO battery charged at 500C and discharged at 50C current rate.

**Table 1 | Comparison with cycling data from literature.**

| | Standard electrochemical measurements | | Non-standard electrode preparation | |
|---|---|---|---|---|
| | Capacity 50C (mAh g$^{-1}$) | Cycles—fading | Capacity 50C (mAh g$^{-1}$) | Cycles—fading |
| Nano-sized hierarchical Li$_{3.5}$Ti$_5$O$_{12}$ (present work) | 170 | 1,000-0% | — | — |
| Nanograins coated by carbon[25] | 128,8 | 500-7.4% (10C) | — | — |
| N-doped 2D wavelike with carbon joints[41] | 151 | 150-6.5% | — | — |
| Assembly on nanowires[42] | — | — | 125 (60C) | 5,000-17% (20C) |
| N-doped, carbon-coated nanocomposite[43] | ≈110 | 300-5% (10C) | — | — |
| Core-shell LTO/C[44] | 101 | 250-11% (1C) | — | — |
| NPs[16] | — | — | 175 | 1,000-11% (100C) |
| NanoLTO—carbon nanotubes composite[17] | ≈130 | — | — | — |

2D, two-dimensional; LTO, Li$_4$Ti$_5$O$_{12}$; NP, nanoparticle.
Present work is summarized in first line, references indicated for previous results from literature.

porous secondary particles, with specific surface area of 220 m$^2$ g$^{-1}$. By a combination of analytical methods (ICP-OES, XPS and XRD), a Li- and O-deficient LTO composition was established. Interestingly, such material delivers exceptional electrochemical performance in lithium-ion batteries, with recorded capacity for low current densities exceeding the theoretical one, as well as reversible capacity of 170 mAh g$^{-1}$ after 1,000 cycles at 50C current density with practically no signs of capacity fading.

## Methods

**Synthesis.** Nanostructured LTO was synthesized using a one pot glycothermal procedure. Typically, 4.59 g (45 mM) of lithium acetate dihydrate was completely dissolved in 200 ml of 1,4-BD under magnetic stirring at room temperature. Then, 17,02 g (50 mM) of titanium (IV) $n$-butoxide was dropwise added and the whole solution was stirred for around one hour at the end of which the reaction medium turned into yellowish transparent solution. Next, the solution was transferred into a 700 ml stainless-steel autoclave. Additional 60 ml of 1,4-BD was added into a gap between the autoclave and the beaker in order to assure thermal contact. The autoclave was tightly sealed and heated up to 300 °C (3 °C min$^{-1}$) and kept at that temperature for 2 h. Solution in autoclave was constantly stirred by mechanical stirrer (300 r.p.m.). The maximum autogenous pressure was obtained at the end of reaction (25 bar). After that, the autoclave was cooled down overnight. White precipitate with milky colloidal solution was obtained. Product was collected by centrifugation (6,000 r.p.m. for 10 min), washed with ethanol three times and then dried in vacuum at 50 °C for 3 h.

**Physico-chemical characterization.** Phase identification of the as-prepared and heat-treated LTO powders was conducted with an X-ray diffractometer (XRD, model: X'Pert Pro from Philips) using CuKα radiation. In addition, Rietveld analysis was performed by using high-resolution XRD data. The morphology and microstructure were studied using a Zeiss Supra 55VP SEM and a Tecnai Osiris TEM. FTIR spectra of the LTO powders were registered using ATR technique within a range of 400–4,000 cm$^{-1}$ using Fourier infrared spectrometer Spectrum 65 made by Perkin Elmer. Thermogravimetric (TGA) TG thermal analyser (Netzsch STA 409 PC). The specific surface area of the prepared sample was

calculated from the adsorption isotherm of nitrogen at 77 K based on the Brunauer–Emmett–Teller method (model: Belsorp max). XPS was performed with a PHI Quantera SXM. The elemental analysis ICP–OES was performed on spectrometer Activa Horiba Jobin Yvon

**Electrochemical characterization.** Electrochemical performances of the synthesized nano LTO were evaluated in coin-type cells with metallic lithium used as the negative electrode. Standard, 1 mol $dm^{-3}$ $LiPF_6$ dissolved in 1:1 vol. ratio of ethylene carbonate and diethyl carbonate was used as the electrolyte. The positive electrode was prepared by mixing of the active material (70 wt.%) with carbon additives (10 wt.% carbon black, 15 wt.% graphite) and polyvinylidene fluoride binder (5 wt.%). N-methyl-2-pyrrolidone was added to the mixture in a certain amount, to obtain the desired viscosity of the paste. After overnight mixing, the slurry was cast on the Al foil, dried under vacuum at 70 °C and then was polymerized at 120 °C.

As the studies presented in this work were directed towards development of an effective, but cheap and scalable way of preparation of the nanosized LTO, also the electrode preparation method and consequently, electrochemical tests were planned in a way to keep the costs minimized. Owing to the use of standard and cheap Al foil, the voltage range was limited to 1.3–2.5 V range. The active material loading was about 1 mg $cm^{-2}$. The cells were constructed using commercial CR2032 enclosures in Ar-filled glove box (with $O_2$ and $H_2O$ levels < 0.1 p.p.m.).

Electrochemical tests (cyclic voltammetry scans, charge/discharge characteristics and AC impedance spectroscopy studies) were conducted using Solartron SI 1287 electrochemical interface and Solartron 1252A frequency response analyser. Electrochemical impedence spectroscopy studies were conducted in 10 mHz–300 kHz range with 25 mV amplitude. Impedance data were fitted using typical equivalent circuit: $R_{ohm} - (R_{dl}CPE_{dl}) - (R_{ct}CPE_{ct})$ where: $R$ stands for resistance and CPE is a constant phase element. The respective indexes correspond to: ohmic-, double layer- and charge transfer-related effects[40,45].

**Data availability.** Data supporting the conclusions presented in this study are available from corresponding author upon request.

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

## Acknowledgements

M.O. was supported primarily by the French Minister of Research through Ecole Normale Superieure and by the LABEX IMUST funding for PhD project.

## Author contributions

M.O. and F.C. designed and synthesized the material, and performed structural characterization. K.Ś., A.R. and D.O. performed electrochemical characterization. M. Ś. performed FTIR spectroscopy. F.L. carried out SEM microscopy together with interpretation of results. S.P. discussed and supervised the work. Manuscript was written by M.O., F.C., K.Ś., F.L. and S.P.

## Additional information

**Competing interests:** The authors declare no competing financial interests.

**Publisher's note**: 

