## [Peer Review File · Nature Communications]

Reviewers' Comments:

Reviewer #2 (Remarks to the Author):

Odziomek et al. reported a method for preparing hierarchically-structured LTO-type nanosized spinel through scalable and facile glycothermal process. The prepared LTO exhibited outstanding electrochemical lithium performances with excellent rate capability and cycle stability. This work is interesting, and I think it could be published in Nature Communications after some major revisions.

1.The reason of the capacity increase in the initial 300 cycles should be explained in detail.

2.The SEM and TEM images of the prepared LTO after 1000 cycles should be provided to further confirm the structure stability.

3.Some literatures regarding LTO should be cited, such as *Electrochimica Acta*, 2016, 211, 119–125; *Advanced Functional Materials* 2014, 24, 4349-4356.

Reviewer #3 (Remarks to the Author):

A. Summary of the key results

The manuscript reports lithium titanate (LTO) nanoparticle-based 3D structure, which can be used as anode in secondary battery. The authors prepared LTO nanostructure by solvothermal method to produce crystalline 4-8 nm-sized LTO, further, and reorganize into self-assembled hierarchical structure (micrometer scale) with some porosity. The nano-sized LTO and porous structure in the self-assembled form is expected to give some benefits in terms of fast transport of Li ion and electrolyte. The electrochemical performance is quite attractive because the reversible capacity at 50 C is 170 mAh/g after 1,000 cycles considering the theoretical capacity of LTO ($\text{Li}_4\text{Ti}_5\text{O}_{12}$, 175 mAh/g). Interestingly, even faster charging at 500C does not give any capacity fading during 100 cycles.

B. Originality and interest: if not novel, please give references

The synthetic approach for 3D LTO nanostructure and characterization is not newly-tried scientific method. However, this paper is interesting and well-organized enough to present sufficient and succinct experiment/result/characterization/electrochemical analysis for the extraordinarily excellent performance. The authors claimed that the ultrafast charging of the synthesized LTO 3D aggregate anode is possible owing to its nano-sized morphology and adequate porous structure. Furthermore, its capacity retention is zero at 50 C during 1,000 cycles.

Fast charging is also achievable at 500 C without capacity fading during 100 cycles. Considering the fact that LTO is known as a zero strain anode material during charging/discharging, all the other references showed somewhat capacity fading for a prolonged cycle (i.e., 1,000 cycles).

C. Data & methodology: validity of approach, quality of data, quality of presentation

This paper presents a solvothermal synthesis of LTO nanoparticle and reorganization into hierarchical structure and utilization as an anode material in a half cell lithium battery. Overall, data acquisition and analysis are on the right track for drawing conclusion. This review is interested in the following questions:

First of all, the LTO nanoparticle formation looks understandable, but it is hard to imagine how the hierarchical-structured LTO aggregates are made into rather spherical shape. This reviewer is wondering whether the authors have tried to change or interrupt the surface stability of the LTO and observe what happens, and finally draw some more convincing conclusion. By doing so, is it possible to control the surface area and pore size of the 3D LTO aggregate?

Secondly, this paper can improve by explaining why lithium- and oxygen-deficient LTO shows excellent rate capability. Why Lithium-deficient phase shows higher initial discharge capacity than the theoretical one at 0.5 C (slow) and, then, decreasing? In contrast, at high C-rate (50 C), the capacity keeps increasing up to a certain level. Plus, charging capacity behavior at low and high rate, looks pretty different. The kinetics on SEI formation seems different. Furthermore, this reviewer is wondering if oxygen-deficient phase has higher electronic conductivity, so it gives better rate capability?

D. Appropriate use of statistics and treatment of uncertainties

The porous hierarchical LTO structure leads to excellent electrochemical performance in terms of rate capability. TEM, BET, phase and composition analyses are sufficient to understand the formation of porous LTO nano-aggregate. Nevertheless, the relationship between lithium-/oxygen-deficient phase and electrochemical performance is not clear.

E. Conclusions: robustness, validity, reliability

The conclusion is well-summarized and succinct in a way that the morphology of the LTO aggregate is relevant to high rate capability. It lacks in explaining the excellent rate capability only with the porous LTO structure.

Suggested improvements: experiments, data for possible revision

It is pretty convincing the LTO aggregate shows excellent rate capability by cycling 1,000 times. The practical use of this material can be further verified by, more, prolong the cycling test, like 10,000 times. Some works have already tested the cycling performance up to 5,000 times and showed a little capacity fading. (e.g., J. Power Source, 2011, 196, 2283)

In figure 2, the schemes in Fig.2C and 2D show two different aggregate formation steps.

However, the size of the aggregates are not that different in C and D, which can be thought of as same aggregates.

F. References: appropriate credit to previous work?

The references cited in the manuscript seems appropriate.

G. Clarity and context: lucidity of abstract/summary, appropriateness of abstract, introduction and conclusions

Line 65 : Check “high volumetric energy density”

Line 197 : Check the sentence to rephrase

Answer to referees (in bold characters in the text):

Reviewer #2 (Remarks to the Author):

Odziomek et al. reported a method for preparing hierarchically-structured LTO-type nanosized spinel through scalable and facile glycothermal process. The prepared LTO exhibited outstanding electrochemical lithium performances with excellent rate capability and cycle stability. This work is interesting, and I think it could be published in Nature Communications after some major revisions.

1.The reason of the capacity increase in the initial 300 cycles should be explained in detail.

This is a very important and interesting point. As it is written in the manuscript, increase of capacity is directly connected to decreasing polarization (line 226-227) and decrease of resistances measured by EIS method (line 253-254). We assigned this phenomenon to rather complicated SEI formation, which is straitened by fast charging and discharging. In order to clarify that in the text we added sentence: "Finally, the increase of capacity in the first 300 cycles at fast charging rate is assigned here to rather complicated formation of SEI, which is additionally straitened by fast charging/discharging rate."

We are not able to explain this phenomenon in detail yet, however we are planning to perform deeper and more sophisticated analysis. We are applying for neutron scattering measurement access at the synchrotron facilities, which is much more sensitive toward Li atoms than XRD. Using this technique, we hope to follow migration of Li ions during charging and discharging cycles. Nevertheless because of very long waiting time for such heavy experiment, the results will be part of a next paper.

2.The SEM and TEM images of the prepared LTO after 1000 cycles should be provided to further confirm the structure stability.

SEM, on the cycled samples, has been performed following the advice of the referee. We compared as prepared anode layer with layer cycled 300 times at 50C and layer cycled 1000 times at 50C plus charged 100 times at 500C and discharged at 50C (Fig. 1). Clearly we can see that porous structure was preserved after 300 and 1000 cycles. Together with microstructure, structure of prepared material did not change as well. XRD diffractogram showed no differences before and after 1000 cycles. We have included these results in the manuscript "Examination of cycled material microstructure by SEM revealed no visible differences. LTO remained highly porous and hierarchically-structured (Supplement Figure S6). Additionally, XRD did not show any

changes in peak broadening. It means that nanoparticles did not change their shape or size.”
and added SEM picture after 300 cycles to supplementary information.

Figure 1 SEM pictures of anode layer prepared from LTO: a) as prepared, b) after 300 cycles at 50C, c) after 1000 cycles at 50C and 100 cycles at charging rate 500C and discharging rate 50C

3. Some literatures regarding LTO should be cited, such as *Electrochimica Acta*, 2016, 211, 119–125; *Advanced Functional Materials* 2014, 24, 4349-4356.

The suggested references were included in the text as references as following :

[24] - *Nanostructured LTO was previously reported in a form of nanosheets [18], nanorods [19], nanotubes [20], nanowires [21], nanoflakes [22] nanoflowers [23] and nanoparticles [24].*

[11] - *At the same time, in order to increase electronic component of the conductivity, various additives (usually carbonaceous materials) are often used, yielding composite-type electrodes [9-11].*

Reviewer #3 (Remarks to the Author):

A. Summary of the key results

☑ The manuscript reports lithium titanate (LTO) nanoparticle-based 3D structure, which can be used as anode in secondary battery. The authors prepared LTO nanostructure by solvothermal method to produce crystalline 4-8 nm-sized LTO, further, and reorganize into self-assembled hierarchical structure (micrometer scale) with some porosity. The nano-sized LTO and porous structure in the self-assembled form is expected to give some benefits in terms of fast transport of Li ion and electrolyte. The electrochemical performance is quite attractive because the reversible capacity at 50 C is 170 mAh/g after 1,000 cycles considering the theoretical capacity of LTO ($\text{Li}_4\text{Ti}_5\text{O}_{12}$, 175 mAh/g). Interestingly, even faster charging at 500C does not give any capacity fading during 100 cycles.

No further comments, the reviewer has pointed out high performance of our material.

B. Originality and interest: if not novel, please give references

☑ The synthetic approach for 3D LTO nanostructure and characterization is not newly-tried scientific method. However, this paper is interesting and well-organized enough to present

sufficient and succinct experiment/result/characterization/electrochemical analysis for the extraordinarily excellent performance. The authors claimed that the ultrafast charging of the synthesized LTO 3D aggregate anode is possible owing to its nano-sized morphology and adequate porous structure. Furthermore, its capacity retention is zero at 50 C during 1,000 cycles. Fast charging is also achievable at 500 C without capacity fading during 100 cycles. Considering the fact that LTO is known as a zero strain anode material during charging/discharging, all the other references showed somewhat capacity fading for a prolonged cycle (i.e., 1,000 cycles).

We are pleased that reviewer notices good organization of our work and above all extraordinarily excellent performance of our material with extreme stability which has never been reported before. Reviewer is right that attempts for synthesis of nanostructure LTO are not new, however for the first time we show possibility of glycothermal synthesis of LTO phase with unusual nanostructuration.

C. Data & methodology: validity of approach, quality of data, quality of presentation

☐ This paper presents a solvothermal synthesis of LTO nanoparticle and reorganization into hierarchical structure and utilization as an anode material in a half cell lithium battery. Overall, data acquisition and analysis are on the right track for drawing conclusion. This review is interested in the following questions:

First of all, the LTO nanoparticle formation looks understandable, but it is hard to imagine how the hierarchical-structured LTO aggregates are made into rather spherical shape. This reviewer is wondering whether the authors have tried to change or interrupt the surface stability of the LTO and observe what happens, and finally draw some more convincing conclusion. By doing so, is it possible to control the surface area and pore size of the 3D LTO aggregate?

Secondly, this paper can improve by explaining why lithium- and oxygen-deficient LTO shows excellent rate capability. Why Lithium-deficient phase shows higher initial discharge capacity than the theoretical one at 0.5 C (slow) and, then, decreasing? In contrast, at high C-rate (50 C), the capacity keeps increasing up to a certain level. Plus, charging capacity behavior at low and high rate, looks pretty different. The kinetics on SEI formation seems different. Furthermore, this reviewer is wondering if oxygen-deficient phase has higher electronic conductivity, so it gives better rate capability?

We are thankful that reviewer confirmed that our analyses are “on the right track for drawing conclusions”.

Reviewer consider formation of nanoparticles as understandable process, nevertheless he wonders how formation of spherical aggregates is possible during this process.

We have added the following sentence and reference in order to clarify it for readers: *Formation of spherical aggregates of particles was previously observed for glycothermal synthesis of YAG nanoparticles where primary particles of the size of few tens of nanometers aggregate into secondary structures. Those primary particles could not be separated once aggregated. However, no further structuration was observed [29].*

(29) Kasuya, R., Isobe, T., Kuma, H., Katano, J. Photoluminescence enhancement of PEG-modified YAG:Ce³⁺ nanocrystal phosphor prepared by glycothermal method. *J. Phys. Chem. B.* **109**, 22126-22130 (2005).

Reviewer also asks if we try to change or interrupt the surface stability of the LTO to control pore size and surface area.

Yes, we synthesized several LTO samples in various conditions (varying precursors, concentration, adding co-solvent and so on) and indeed we observed minor or major changes in surface area (200-350 m³/g) and pore size distribution (fig. 2 just for referee). We characterized one compound from the electrochemical point of view. Nevertheless, we are going to investigate the electrochemical properties with varying hierarchical structuration and this will be the purpose of a next article.

Figure 2 Nitrogen adsorption and desorption isotherms obtained for LTO synthesized in different conditions.

Reviewer asks for reason of higher initial discharge capacity at low current in lithium-deficient phase.

We mentioned in the paper line from 202 to 206: *While on first discharge such behaviour might be interpreted as originating from initial introduction of missing lithium, speculatively into the 8a site with corresponding 29 mAh g⁻¹ of additional capacity, the enhanced capacity on the following cycles should be rather linked with near-surface lithium storage, with possible role of the surface-present organics originating from the synthesis process [35, 36].*

We rephrased it in order to make it more clear:

While on first discharge such behavior might be interpreted as originating from initial introduction of missing lithium, speculatively into the 8a site with corresponding 29 mAh g⁻¹ of additional capacity, the enhanced capacity on the following cycles is possibly linked with near-surface lithium storage, with contributions of the surface-present organics originating from the synthesis process [38, 39].

Difference between battery behavior during fast and slow cycling are substantial question as reviewer noticed. It is likely that surface-related storage of lithium is more dominant for higher rates, while bulk one is more pronounced at lower current. Consequently, the observed characteristics may differ considerably, as in such fine powder surface atoms comprise a large part of the nano-grains. However, as in the case of the first reviewer question, we still do not have complete answer on that topic but we will try to reveal it in the next paper.

Oxygen-deficient phases could increase the electrical conductivity of LTO as it was already reported, however this electrical conductivity enhancement is mostly caused by existence of Ti^{3+} , which are not observed in our case. It is not straightforward to say whether the electronic conductivity would be improved or not. However, due to nano-crystallinity of the powder, discussion about charge state of the atoms and transport properties is also hindered. It would be of interest to calculate and measure energy gap for the LTO as a function of the grain size (vide quantum dots), but this is not within the scope of this paper.

Speculatively, if missing Li ions are indeed incorporated in the LTO structure, then in order to compensate charge Ti would need to reduce its oxidation state to 3+ because there is no source of oxygen to incorporate in the structure. If this is the case it can easily explain the different kinetics in fast and low charging. During fast charging Li is not completely incorporated in the structure due to SEI formation and lack of time so gradual feeling of missing Li ions would improve electrical conductivity during ongoing process. In case of slow charging Li have enough time to accommodate correct position in the structure.

D. Appropriate use of statistics and treatment of uncertainties

☒ The porous hierarchical LTO structure leads to excellent electrochemical performance in terms of rate capability. TEM, BET, phase and composition analyses are sufficient to understand the formation of porous LTO nano-aggregate. Nevertheless, the relationship between lithium-/oxygen-deficient phase and electrochemical performance is not clear.

Reviewer points out sufficient number of analytical technics to understand formation of porous phase, which is directly connected with excellent electrochemical properties of material. As we discussed before, the relationship between lithium-/oxygen-deficient phase and extraordinary electrochemical performance will be studied at synchrotron facilities and reported in a next paper.

E. Conclusions: robustness, validity, reliability

☒ The conclusion is well-summarized and succinct in a way that the morphology of the LTO aggregate is relevant to high rate capability. It lacks in explaining the excellent rate capability only with the porous LTO structure.

Suggested improvements: experiments, data for possible revision

☒ It is pretty convincing the LTO aggregate shows excellent rate capability by cycling 1,000 times. The practical use of this material can be further verified by, more, prolong the cycling test, like 10,000 times. Some works have already tested the cycling performance up to 5,000

times and showed a little capacity fading. (e.g., J. Power Source, 2011, 196, 2283)
In figure 2, the schemes in Fig.2C and 2D show two different aggregate formation steps. However, the size of the aggregates are not that different in C and D, which can be thought of as same aggregates.

We are pleased that reviewer found our work as well summarized and concluded, and that he shares our opinion of relevant morphology of LTO to high rate capability. However, the reviewer writes that porous structure is not enough to explain excellent electrochemical properties. We completely agree with this statement. Another factor, which could have influence on material performance is very high crystallinity of LTO despite of small crystallites size. In order to clarify that in the text we modify the sentence: *Doubtlessly, such the extraordinary results can be connected with high ability of Li ions to be inserted or extracted into/from nanoparticles with well-developed surface [36].*

to:

Doubtlessly, such extraordinary results are connected with high ability of Li ions to be inserted or extracted into/from nanoparticles with well-developed surface [39] and with peculiar non-stoichiometry of the system. Additionally, very high crystallinity is also an important factor to be considered for good performances.

Reviewer points that material can be further investigated at even larger number of cycles (10 000 time). We plan to include it in the next paper, however 1000 cycles is already very high number of cycles, only few papers exceeded that number. Some of them were cited by us [3, 43]

Reviewer considers that picture 2c and 2d shows two different aggregate formation steps. Indeed, in the picture 2d we show micron sized arrangements of nano-aggregates that are showed in the picture 2c.

F. References: appropriate credit to previous work?

The references cited in the manuscript seems appropriate.

G. Clarity and context: lucidity of abstract/summary, appropriateness of abstract, introduction and conclusions

Line 65 : Check “high volumetric energy density”

Line 197 : Check the sentence to rephrase

We do not find any confusion by using “high volumetric energy density” expression. Volumetric energy density is commonly used unit by electrochemists.

We rewritten sentence in the line 197:

Interestingly, the recorded at low current rate of C/2 curves exhibit a substantial (c.a. 30-40 mAh g⁻¹) capacity range with voltage constantly changing with the ongoing electrode process, which is further followed by a more flat dependence, and finally, larger changes at the final stages of the charge/discharge (Figure 3a).

To:

Interestingly, the recorded curves, at low current rate of $C/2$, show a substantial capacity range (c.a. 30-40 mAh g⁻¹) with voltage constantly changing with the ongoing electrode process. This is further followed by a more flat dependence, and finally, larger changes at the final stages of the charge/discharge (Fig. 3a).

Reviewers' Comments:

Reviewer #2 (Remarks to the Author):

The manuscript has been revised accordingly as the reviewers suggested, and it should be published.

Reviewer #3 (Remarks to the Author):

This interesting paper is ready to publish. Overall, the authors made an effort to fulfill the reviewers' comments by replying point by point, and the revised article seems sufficient to publish in Nature Communication.